

Reviews and Syntheses: Understanding the impacts of peatland catchment
management on DOM concentration and treatability
Jennifer Williamson[1][*], Chris Evans[1], Bryan Spears[2], Amy Pickard[2], Pippa J. Chapman[3], Heidrun
Feuchtmayr[4], Fraser Leith[5], Susan Waldron[6], Don Monteith[4]
[1]UK Centre for Ecology & Hydrology, Environment Centre Wales, Deiniol Road, Bangor, Gwynedd, LL57 2UW
[2]UK Centre for Ecology and Hydrology, Bush Estate, Penicuik, Midlothian, EH26 0QB
[3] School of Geography, Faculty of Environment, University of Leeds, Leeds, LS2 9JT
[4]UK Centre for Ecology & Hydrology, Lancaster Environment Centre, Library Avenue, Bailrigg, Lancaster, LA1 4AP
[5]Scottish Water, 6 Castle Drive, Dunfermline, KY11 8GG
[6]School of Geographical and Earth Sciences, University of Glasgow, Glasgow G12 8QQ
[*]*Corresponding Author* (jwl@ceh.ac.uk)





**Abstract**
In the UK most large reservoirs constructed for public water supply are in upland areas and situated
in catchments that contain at least some organic-rich soils. Dissolved organic matter (DOM) leaching
from these soils imparts a brownish colour to water and raises treatment challenges for the water
industry since excessive post-treatment concentrations result in the generation of potentially harmful
disinfection by-products in drinking water. The primary method for maintaining sufficiently low pre-
disinfection DOM concentrations is chemical coagulation, but in the past 15 years water companies
have increasingly considered the capacity for catchment interventions to improve raw water quality
at source, reducing the need for costly and complex engineering solutions in treatment works. There
remains considerable uncertainty around the effectiveness of these catchment engineering-based
measures and a comprehensive overview of the research in this area remains lacking. Here we review
the peer-reviewed evidence for the effectiveness of four management options for upland organic soil-
dominated catchments that are being considered by the water industry as options for controlling DOM
releases. These are ditch blocking, revegetation, reducing forest cover, and cessation of managed
burning. Results of plot scale investigations into effects of ditch blocking on ditch-blocking are
available but largely equivocal, while there is a paucity of information regarding impacts at spatial
scales of more direct relevance to water managers. 'The presence of plantation forestry on peat soils
is generally associated with increasing DOM concentrations, although canopy removal has little short-
term benefit and can even further increase concentrations. Although not widely studied, the available
evidence suggests that *Sphagnum* mosses produce DOM that is more easily removed via conventional
treatment processes compared to vascular plants such as heather and grass species. We found
surprisingly little published research around the extent to which manipulation of in-reservoir
processes might be used to mitigate or exacerbate changes in inflowing DOM as part of a catchment
management approach.
This review concluded that catchment management measures have rarely been monitored with
downstream water quality as the focus, and that restoration impacts vary across sites. To mitigate the
uncertainty surrounding restoration effects on DOM, measures should be undertaken on a site-
specific basis, where the scale, effect size and duration of the intervention are considered in relation
to subsequent biogeochemical processing that occurs in the reservoir, the treatment capacity of the
water treatment works and future projected DOM trends.




## Introduction

Peatland restoration has become an integral part of the UK environment strategy, particularly in the drive toward Net Zero. It is founded on the potential to achieve multiple benefits that include improving biodiversity, enhancing carbon sequestration, and controlling water runoff and quality, in catchments that are deemed to have been degraded by anthropogenic stressors. Nearly three quarters of the storage capacity of drinking water reservoirs in the UK is sourced from peatland areas (Xu et al., 2018). The dissolved organic matter DOM concentrations of these water tend to be relatively high, and have been rising since the 1980s (e.g. Naden and Mcdonald, 1989; Robson and Neal, 1996; Harriman et al., 2001; Freeman et al., 2001; Worrall et al., 2004). Mean DOM concentrations in UK Upland Waters Monitoring Network (UWMN) surface waters, most of which are dominated by organic-rich soils, have approximately doubled over the last three decades being approximately double those seen in the late 1980s (Figure 1). At the sub-catchment scale, Chapman et al. (2010) found that water colour increased by between 22 and 155 percent over a 20 year period between 1986 and 2006. This phenomenon has now been observed across much of industrialised North America and north-west Europe, and appears to largely result from an long-term increase in the solubility of terrestrial organic matter as soils recover from the effects of acid rain (Monteith et al., 2007; De Wit et al., 2021). Rising levels of DOM in waters draining many of these catchments pose considerable water treatment challenges, with respect to increasing treatment costs and risks of regulatory failure (see Figure 1).  It has been proposed that peatland restoration measures might help slow or even reverse these DOM trends, but while some of the benefits of peatland restoration are now becoming clear (e.g. Glenk and Martin-Ortega, 2018), evidence for impacts on water quality have been more difficult to glean.

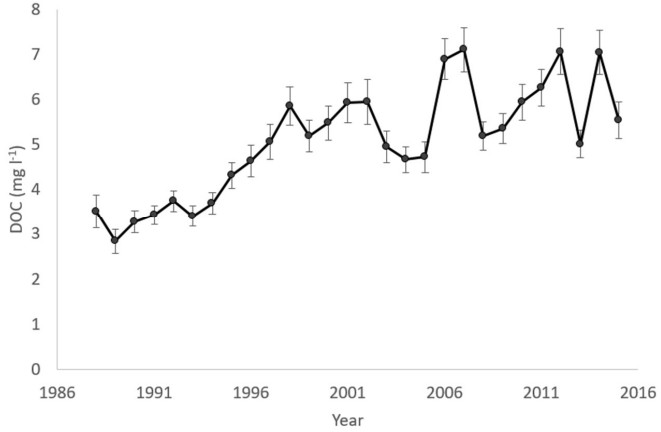

Figure 1: Mean (+/- Standard error) annual dissolved organic carbon (DOC) concentrations from UWMN sites. These sites are predominately situated in the north and west of the UK – see www.uwmn.uk for more details.

Although consumption of DOM in drinking water is not directly harmful to people, coloured water reduces customer satisfaction (Ritson et al., 2014) and can be indicative of further problems. Indirectly, elevated DOM concentrations have implications for human health due to their potential



influence on treatment processes and the production of carcinogenic disinfectant by-products (DBPs) such as trihalomethanes (THMs), which are regulated by the Drinking Water Inspectorate (DWI) due to their potential carcinogenic properties. Chlorination, a standard disinfection process in most UK WTWs, leaves free chlorine in the water supply as a residual disinfectant. Free chlorine reacts with DOM remaining in the water supply following coagulation and filtration to form DBPs, including THMs. Chloramination, the treatment of drinking water with chlorine and ammonia to form chloramine, has been used as a method of reducing THM formation. However, it has been found that chloramination promotes the formation of nitrogenous DBPs (e.g. Bond et al., 2011; Lavonen et al., 2013), which are more carcinogenic than THMs (Ding and Chu, 2017) and are likely to be regulated in the future. DOM also may hamper the efficacy of chlorine as a disinfectant while simultaneously acting as a substrate for bacterial regrowth (Prest et al., 2016), thus increasing the risk of regulatory failure from bacterial contamination and the subsequent loss of customer trust.

The composition of DOM can have a large influence on the performance of the water treatment processes and the formation of DBPs upon chlorination (Matilainen et al., 2010). DOM in water draining peatland areas tends to be predominantly hydrophobic, and relatively photoreactive and biologically recalcitrant. It is relatively easily removed by conventional coagulation and filtration during drinking water treatment due to the presence of charged functional groups (Matilainen et al., 2010). Hydrophilic DOM, on the other hand, is mostly produced within the waterbodies by phytoplankton activity (Imai et al., 2002), and is biologically labile but less easily degraded by sunlight (Berggren and Del Giorgio, 2015; Berggren et al., 2018). The relative balance of hydrophobic to hydrophilic DOM in water is referred to as hydrophobicity, and is conventionally assessed in the water treatment system using Specific UV Absorbance measurements at 254 nm (SUVA$_{254}$), i.e. absorbance at 254 nm per unit dissolved organic carbon concentration (Weishaar et al., 2003). Values greater than 4 indicate hydrophobic dominance, while values less than 2 show the DOM is primarily hydrophilic and will not be effectively removed using conventional coagulation and filtration alone (Matilainen et al., 2010).

Higher concentrations of DOM in raw water necessitate a greater amount of treatment to provide potable water to customers (Monteith et al., 2021). This may include larger coagulant dosages, shorter filter run times, and longer and more frequent cleaning of filtration units, and result in higher energy costs, higher sludge removal costs and an increase in direct and indirect (energy-related) greenhouse gas (GHG) emissions from the treatment process (Jones et al., 2016). Overall, the cost of DOM removal in UK water supplies is estimated to be hundreds of millions of pounds, and has risen sharply in recent years as a direct consequence of rising DOM concentrations. Major additional costs are incurred where capital investment is needed to upgrade treatment infrastructure designed for lower concentration ranges experienced in the past. It is important, therefore, for water industry decision makers to understand the extent to which peatland restoration could make a positive contribution to reducing DOM concentrations of raw water and thus relieve stresses on the treatment system and potentially remove the need for major additional capital investment in treatment plant. This work reviews the available peer-reviewed literature and provides a qualitative assessment of the impacts of peatland restoration on DOM concentrations and treatability.

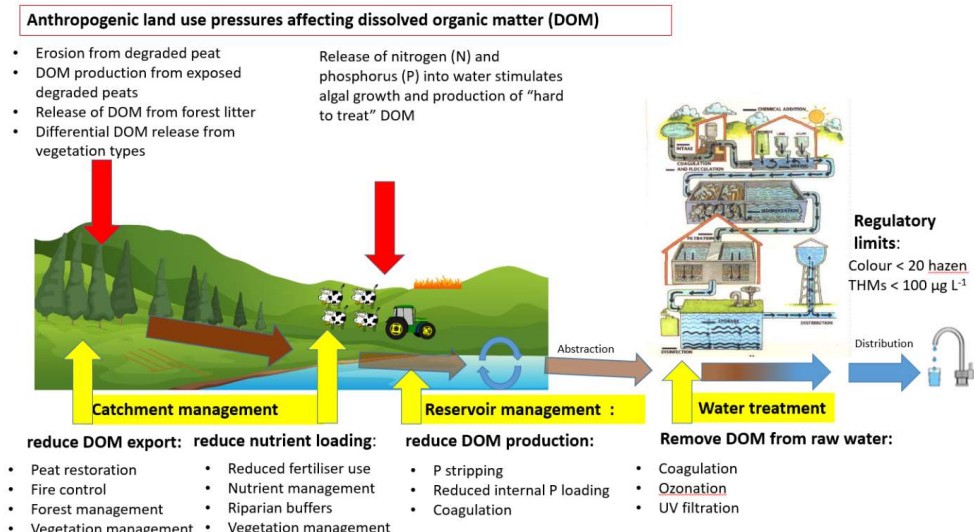

Figure 2: Schematic showing anthropogenic pressures on peatland catchments, and the potential peatland management processes covered in this review.

## 2. Evidence for the efficacy of catchment management approaches in the reduction of DOM

### 2.1. Ditch blocking

Following peatland drainage, the resulting reductions in water tables, loss of peat forming plant species, and consequent drying and cracking of peat surfaces, exposed previously permanently saturated organic matter to oxidative processes, making it more vulnerable to erosion and dissolution into DOM (e.g. Clark et al., 2009). Extensive efforts have been made by the water industry and organisations concerned with peatland conservation to block ditches in an attempt to restore the hydrological, biogeochemical and ecological functions of these landscapes (Figure 3).

Of the studies relevant to UK peatlands found during this review, four out of five (Table 1) reported significant changes in DOM concentrations within peat soil pore water (i.e. plot scale) following ditch blocking, with a cross-study average 34% reduction (range 0 to 69%) (Wallage et al., 2006; Holl et al., 2009; Haapalehto et al., 2014; Strack et al., 2015; Menberu et al., 2017). While therefore suggesting a general tendency for ditch blocking to reduce pore water DOM concentrations, these studies do not necessarily imply that effects will be translated through to surface waters and ultimately to the point of abstraction.

At the ditch scale, results are more variable than those for pore waters (Table 1). The ten studies reviewed showed a mean 10% increase in DOM concentrations following ditch blocking, although this figure is skewed by the large increases reported by Worrall et al. (2007b) and Haapalehto et al. (2014) (100% and 50-75% increases respectively); the median change is zero. Importantly, no significant change in DOM concentration was reported in half of these studies (O'brien et al., 2008; Gibson et al., 2009; Armstrong et al., 2010; Wilson et al., 2011; Evans et al., 2018). Likewise, a recent study monitoring DOM concentrations six years after ditch blocking on a blanket bog found no reduction in DOM concentrations in the restored site compared to the ditched site (and both drained and restored site DOM concentrations remained elevated compared to the non-drained control (Pickard et al.,



2022). Differences in apparent effect size may be related to experimental design, including whether
the work included a simultaneous control and the time period over which post-restoration monitoring
was carried out.
Measuring and reporting water fluxes (and hence DOM fluxes) at a site- or catchment-scale requires
careful consideration of the potential for dominant water flow pathways to be altered following ditch
blocking. For example, Holden et al. (2017) showed that damming of drainage ditches in North Wales
did reduce discharge along the original ditch lines following blanket bog re-wetting, but that most, or
all, of the displaced flow instead left the peatland via overland flow or near-surface through-flow.
Subsequent reporting from the same experiment demonstrated that DOM concentrations in water
displaced along these surficial pathways were approximately the same as those in water travelling
along the ditches, with the result that ditch-blocking was not found to have any clear effect on either
DOM concentrations or fluxes at the catchment scale (Evans et al., 2018). Studies of DOM flux changes
following ditch blocking report an average 24% reduction (range 0 – 88% reduction) in DOM flux,
primarily attributed to decreased water fluxes from the restoration site.
**Table 1: Summary of the impacts of drainage ditch blocking on DOM concentrations and fluxes from peatlands, reported**
**in increasing time since ditch blocking. BA = Before/After, CI = Control/Intervention**

| Reference | Location | Sampling scale | Concentration or flux measured | Time since ditch blocking | Experimental Design | Change since ditch blocking |
|---|---|---|---|---|---|---|
| Worrall et al. (2007b) | UK, blanket bog | Ditches | DOM concentration | 7 months | BACI | 100% increase in DOM concentration. |
| Turner et al. (2013) | UK, blanket bog | 0 and 1st order ditches | DOM concentration and flux | 1 year | BACI | DOM concentration decreased by 2.5% compared to control, DOM flux decreased by 2.2 – 9.2% as a result of decreased water export. |
| Gibson et al. (2009) | UK, blanket bog | Ditches | DOM concentration and flux | 1 year | CI | DOM concentrations unchanged, water flux decreased by 39% meaning DOM flux also declined by the same amount. |
| Wilson et al. (2011) | UK, blanket bog | Ditches and headwater streams | DOM concentration and flux | 2 years | BACI | DOM concentrations unchanged, fluxes were 88% lower in streams draining ditch-blocked catchments due to much lower estimated water export. |
| O'brien et al. (2008) | UK, blanket bog | Headwater streams | DOM flux and water colour | 2 years | BACI | Water colour was unchanged. Fluxes decreased by 24% in streams as a result of decreasing water export. |
| Menberu et al. (2017) | Finland fen, pine mire and spruce mire | Pore water | DOM concentration | 3 years | BACI | 41% reduction in DOM concentration. |
| Evans et al. (2018) | UK, blanket bog | Ditches | DOM concentration | 4 years | BACI | No change in DOM concentration |
| Wallage et al. (2006) | UK, blanket bog | Pore water | DOM concentration | 5 years | CI | DOM concentration lower in porewaters adjacent to blocked ditches (69% lower compared to open ditches) |



| Haapalehto et al. (2014) | Finland, raised bog | Pore water | DOM concentration | 5 years and 10 years | Chronosequence | DOM concentration approx. 10% lower in sites 5 years post restoration and 25% lower in sites 10 years post restoration |
|---|---|---|---|---|---|---|
| Haapalehto et al. (2014) | Finland, raised bog | Ditches | DOM concentration | 5 years and 10 years | Chronosequence | Concentrations approx. 75% higher in sites 5 years post restoration and 50% higher in sites 10 years post restoration |
| Armstrong et al. (2010) | UK, blanket bog | Ditches | DOM flux | 7 years | CI | No change in DOM flux |
| Strack et al. (2015) | Canada, bog | Pore water and ditch water | DOM concentration | 10 years | CI | No change in pore water DOM concentration. Ditch water DOM concentrations were similar in spring and summer and up to 30% lower in the restored site in autumn. |
| Armstrong et al. (2010) | UK, blanket bog | Ditches from a survey in Northern England and Northern Scotland | DOM concentration | 6 months to 18 years | Survey | DOM concentrations 28% lower on average in blocked drains compared to unblocked drains. |
| Holl et al. (2009) | Germany, ex-fenland extraction site | Pore water | DOM concentration | 20 years | CI | DOM concentrations 37% lower at restored site compared to drained site. |
| Urbanova et al. (2011) | Czech Republic, bog | Pore water | DOM concentration | NA comparison between drained and intact sites | CI | No difference in DOM concentration between intact and moderately degraded site, 50% higher DOM concentrations at highly degraded site. |
| Pickard et al. (2022) | UK, blanket bog | Headwater streams | DOM concentration | 6-8 years | CI | No difference in DOM concentration between drained and restored sites. DOM concentrations significantly higher (50% increase) in drained and restored sites compared to non-drained controls. |



Nine studies to date have assessed the potential impact of ditch blocking on DOM treatability and
hence the ease of treatability within a conventional water treatment works. They found that the
majority of studies at UK and continental European ditch blocking locations, along with results from
their experimental work, showed little effect of ditch blocking on DOM treatability as measured by
commonly reported metrics such as SUVA, E2:E3 ratios (ratio of light absorbance at 250 and 365 nm)
and E4:E6 ratios (ratio of light absorbance at 465 and 665 nm) (Glatzel et al., 2003; Strack et al., 2015;
Gough et al., 2016; Lundin et al., 2017; Peacock et al., 2018). While none of the studies included direct
measures of DOM hydrophobic and hydrophilic fractions, one measured THM formation potential and


found no change between water samples taken from drained and rewetted blanket bog mesocosms
(Gough et al., 2016), suggesting that in the short term ditch blocking may not reduce THM formation
following water treatment.
More broadly, therefore, while the evidence suggests that ditch blocking may reduce DOM
concentrations within pore waters (Table 3), there is no published evidence for such activities to have
successfully influenced DOM concentrations in runoff at a catchment scale, and thus at a level of
potential relevance to raw water supply to treatment works. It is important to note, however, that
catchment-scale studies are hugely challenging logistically and financially to design and maintain and
are currently very rare over timescales suitable to detect land management effects on water quality.

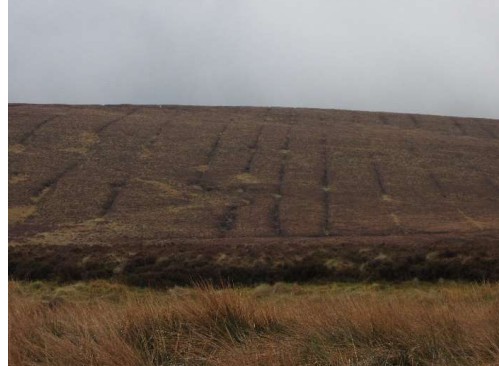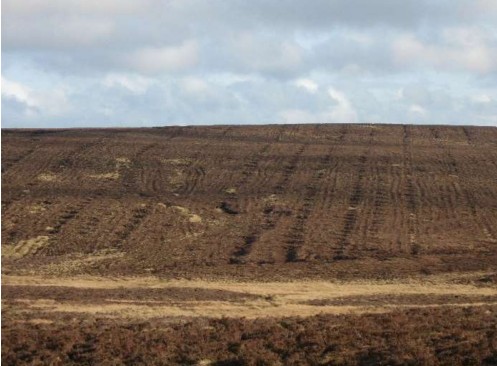


**Figure 3: Drainage ditches before (left) and after (right) blocking on a blanket bog in North Wales, the ditches run down**
**the slope and individual dams can be seen crossing the ditches (Photos: Chris Evans).**

### 2.2. Re-vegetation of bare peat

Exposure of bare peat following anthropogenic disturbance has been an extensive problem in a
number of UK peatland regions, most notably in the Peak District. The subsequent erosion of the peat
has caused significant problems for the water industry because of the high particulate loads from the
catchment to the downstream reservoirs. There have been significant efforts in recent years to
revegetate some of the most degraded upland peatland areas in order to stabilise these systems.
Published research on the impacts of revegetation of peatland areas on DOM is limited, but Qassim
et al. (2014) found that pore water DOM concentrations were higher in revegetated sites compared
to bare peat areas and vegetated controls over a five-year period. The initial revegetation mix in this
work was a nurse crop of *Agrostis* sp., *Deschampsia flexuosa* and *Festuca* sp. in combination with
additions of lime and fertiliser to ensure grass growth. Heather brash was applied to stabilise the peat
surface and provide a seed source of peatland species. The use of lime is likely to have increased DOM
solubility through a reduction in acidity of the peat (Evans et al., 2012), and the re-establishment of
vegetation may have increased the production of 'new' DOM via root leachate and fresh litter
decomposition. Particulate losses from peatland systems decreased following stabilisation of the peat
surface through revegetation irrespective of gully blocking activities (Pilkington et al., 2015), as
overland flow velocities are lower on vegetated peat than bare peat (Holden et al., 2008). However,
the same study (Pilkington et al., 2015), and more recent assessments of the effects of revegetation



on DOM concentrations (Stimson et al., 2017; Alderson et al., 2019), found no long-term changes in
DOM concentrations following revegetation at the headwater catchment scale.
Radiocarbon ($^{14}$C) measurements of DOM in UK upland waters indicate that the principal source of
DOM in waters draining relatively undisturbed soils is recent primary production, probably formed
within the last few years (Evans et al., 2014). It follows, therefore, that plant productivity, and plant
tissue composition and degradability, which depend both on ambient environmental conditions and
species composition, may be important factors, both for DOM concentrations and the treatability of
the DOM produced. In a laboratory-based extraction experiment DOM leached from *Sphagnum* was
more easily removed by a conventional coagulation process and decomposed more rapidly than DOM
leached from *Molinia caerulea* or *Calluna vulgaris* litter. In addition, *M. caerulea* and *C. vulgaris* litter
released more DOM per unit dry weight compared to Sphagnum litter (Ritson et al., 2016). At the field
scale published results are less clear cut: one study found that DOM concentrations in pore waters
were higher in areas of blanket bog dominated by *C. vulgaris* compared to areas dominated by sedges
or Sphagnum species (Armstrong et al., 2012). In contrast, Parry et al. (2015) found no correlation
between dominant vegetation type (differentiated into ericoid, grasses, sedges and bare peat) and
stream water DOM concentrations in headwater catchments. This may reflect the greater
heterogeneity of peatland environments at the catchment scale in comparison to single species
experiments.
The evidence available to date suggests that while revegetation of peatland sites has stabilised bare
peat surfaces (e.g. Pilkington et al., 2015), and is likely to have reduced particulate organic matter loss,
it has not changed DOM export from peat headwater catchments. Laboratory based work has shown
that the species present could impact DOM treatability, with *Sphagnum* derived DOM being more
easily treatable that *M. caerulea* or *C. vulgaris* litter (Ritson et al., 2016). This suggests that catchment
management via revegetation should aim to achieve high cover of *Sphagnum* species compared to
vascular plants to maximise DOM treatability (Table 3). However, as with other restoration measures
there is currently little in the peer-reviewed literature to demonstrate the effectiveness of this at a
catchment scale.
**2.3. Plantation forestry / deforestation**
It has long been recognised that forestry activities can have detrimental impacts on reservoir water
quality and treatability. For example, in 1984 it was shown that drainage and deforestation resulted
in large sedimentation issues at Crai Reservoir in south Wales (Stretton, 1984 cited in: Hudson et al.
1997). Large pulses of nutrients (N and P) can also occur after forest-felling (Neal, 2002).
To reduce the impacts of forest operations on sediment and nutrient loss and consequent raw water
quality in the UK, the Forest and Water Guidelines now state that no more than 20% of a drinking
water catchment should be felled in any 3 year period (Forestry Commission, 2017). In addition to
this, although primarily to conserve soil carbon stocks rather than for improved water quality, the
2000 Forestry Commission guidance note on forest and peatland habitats (Patterson and Anderson,
2000) states that approval will no longer be given for forestry planting or regeneration on active raised
bog or inactive raised bogs that could be restored to active bog, and areas of active blanket bog greater
than 25 ha area and > 45 – 50 cm depth.
A recent review for Yorkshire Water (Chapman et al., 2017) noted that conventional conifer site
preparation on peat, peaty gley and peaty podzol soils would be expected to increase DOM
concentrations. This would be largely due to the implemented drainage reducing the height of the
water table and consequently increasing the production of DOM via increased aeration of the peat





surface (Clark et al., 2009). Jandl et al. (2007), in their review of studies of the effect of forest
management on soil carbon sequestration, highlighted two Finnish studies where DOM
concentrations increased following drainage ditch installation but returned to pre-drainage levels later
in the forest cycle, while Schelker et al. (2012) observed increased colour in sites being prepared for
forestry in northern Sweden. Furthermore, Rask et al. (1998) reported an increase in colour in streams
draining peat dominated catchments following afforestation in Finland, while in Sweden afforestation
has also been linked to long-term increases in water colour (Skerlep et al., 2019). At a regional to
national scale in the UK recent work suggests that the presence of plantation forestry on peat soils
increases DOM concentrations in streams and rivers compared to peat soils with semi-natural
vegetation (Williamson et al., 2021).

Table 2: UK studies reporting DOM concentration monitoring of forestry activities on peat. Note that
where percentage differences are preceded by ~ concentrations were not explicitly listed in text,
figures and tables or supplementary information so are estimated from graphs.

| Paper | Location | Forestry activity monitored | Scale | % difference |
|---|---|---|---|---|
| Muller and Tankere-Muller (2012) | Flow Country | Felling compared to blanket bog | Stream (upstream and downstream) | -6% |
| Zheng et al. (2018) | Central Scotland | Felling compared to windfarm on blanket bog | Stream | ~ 100% |
| Muller et al. (2015) | Flow Country | Felling compared to blanket bog | Stream | No difference |
| Shah and Nisbet (2019) | Central Scotland (raised bog) | Before / after felling | Stream | 0%, 29% & 51% (mean 27%) |
| Cummins and Farrell (2003) | Ireland | Before / after felling | Stream | ~0 – 100% |
| Gaffney et al. (2020) | Flow Country | Before / after felling and felling compared to blanket bog | Stream | No significant difference |
| Muller et al. (2015) | Flow Country | Before / after felling | Ditch | ~ 75% |
| Gaffney et al. (2018) | Flow Country | Before / after felling | Ditch | ~ 150% |
| Cummins and Farrell (2003) | Ireland | Before / after felling | Ditch | ~50% |
| Gaffney et al. (2018) | Flow Country | Felling compared to blanket bog | Ditch | ~500% |
| Muller and Tankere-Muller (2012) | Flow Country | Felling compared to blanket bog | Ditch | 30-325% (overall average 159%) |
|  |  |  |  |  |
| Gough et al. (2012) | North Wales | Presence / absence of forestry | Pore waters | -19% - 111% (average 45%) |
| Howson et al. (2021) | Flow Country | Presence / absence of forestry | Pore waters | ~ 66% |



| Howson et al. (2021) | Central Scotland (raised bog) | Presence / absence of forestry | Pore waters | ~14% |
|---|---|---|---|---|
| Flynn et al. (2022) | Ireland | Presence / absence of forestry | Pore waters | ~400% |
| Gaffney et al. (2018) | Flow Country | Presence / absence of forestry | Ditch | ~ 100% |
| Flynn et al. (2022) | Ireland | Presence / absence of forestry | Stream | No significant difference |
| Shah et al. (2021) | Flow Country | Presence / absence of forestry – time series | Stream | No significant difference |
| Cummins and Farrell (2003) | Ireland | Presence / absence of forestry | Stream | ~25% |


The presence of forestry on peat soils in a UK and Irish context is associated with higher pore water
DOM concentrations across the four studies covered in this review (Table 2), with a mean difference
of approximately 130%. The exception to this pattern was found in spruce plantations in north Wales
where DOM concentrations in pore waters were 19% lower than in adjacent blanket bog (Gough et
al., 2012). We found only one study (Gaffney et al., 2018) comparing DOM concentrations at a ditch
scale between forested and intact blanket bog areas, with DOM concentrations being approximately
100% higher in ditches draining the forested areas. At the stream scale the presence of forestry on
peat had less clear cut impacts on DOM concentrations, with two out of three studies reporting no
significant difference between streams draining catchments with forestry and intact blanket bogs
(Shah et al., 2021; Flynn et al., 2022), and the third showing an DOM concentrations approximately
25% higher in a stream draining a forested catchment compared to a blanket bog catchment (Cummins
and Farrell, 2003).
Tree felling tends to produce larger increases in DOM, though the effects are not universal across
studies and locations. At the stream scale three of five studies reported increases following felling
(Cummins and Farrell, 2003; Zheng et al., 2018; Shah and Nisbet, 2019), with a mean increase of
approximately 43%, although the two studies in the Thurso catchment showed no change (Muller et
al., 2015) and a 6% decrease in concentrations (Muller and Tankere-Muller, 2012), which was
attributed to the success of buffer strips between the plantation and the monitored stream. At the
ditch scale the mean increase in DOM concentrations was nearly 200% (ranging from a 50% increase
to a 500% increase, see Table 2) (Cummins and Farrell, 2003; Muller and Tankere-Muller, 2012; Muller
et al., 2015; Gaffney et al., 2018).
There has been comparatively little research on the effects of forest presence on the treatability of
DOM, although Gough et al. (2012) evaluated DOM concentrations and $SUVA_{254}$ values in waters
draining catchments forested with different tree species. They found that pore water leachates from
pine and larch plantation yielded particularly high DOM concentrations relative to a blanket bog
control (19 and 13 mg L$^{-1}$, respectively, compared to 9 mg L$^{-1}$). Leachates also had lower $SUVA_{254}$ values
(1.2 and 2.4 respectively, compared to 3.3 L mg$^{-1}$ m$^{-1}$). This would suggest that DOM leaching from
plantations dominated by these tree types may be less easily treatable than DOM from blanket bogs.
Similarly, samples taken from Scottish blanket and raised bog sites (Howson et al., 2021) found that
$SUVA_{254}$ values were lower from forested sites, again suggesting that forestry on peat results in less
aromatic, hydrophobic DOM that may be less easily removed via conventional coagulation.



Recently there have been attempts to restore previously afforested fen and bog peatlands in parts of
Europe and North America under what is often referred to as 'forest-to-bog' restoration (Chimner et
al., 2017; Andersen et al., 2017). Although still a relatively new practice within the UK, this type of
restoration has been carried out for 18 years in the Flow Country in northern Scotland, and national
policies on peat restoration may lead to its expansion in the future. Some of the studies listed in Table
2 (Muller and Tankere-Muller, 2012; Muller et al., 2015; Gaffney et al., 2018; Shah and Nisbet, 2019;
Gaffney et al., 2020; Howson et al., 2021; Shah et al., 2021) monitored the impacts of felling as part
of ongoing forest-to-bog restoration monitoring, with the main differences in management being that
the trees were felled to waste (the practice of leaving felled trees *in-situ* to rot) and there was less
ground disturbance at the site compared with the use of machinery to extract felled timber (Gaffney,
2017). However, the practice of felling trees to waste has been suggested to provide a potential
additional DOM source as the trees slowly decompose (Muller et al., 2015), with mulched fallen trees
providing a major source of water soluble DOM (Howson et al., 2021).
As bog vegetation regenerated in the Flow Country, DOM concentrations reduced from elevated levels
towards those seen in forest control areas, although the time frame for complete recovery to pre-
intervention levels is to date inconsistent, with some areas showing elevated DOM in the restoration
sites compared to the control sites after 17 years (Gaffney et al., 2018). However, in others DOM
concentrations had returned to those seen in intact blanket bog within the same time frame (Howson
et al., 2021), or were showing inconsistent effects across sub-catchments, with the most upstream
catchments showing increased DOM concentrations compared to bog controls, an effect not seen
further downstream (Pickard et al., 2022). Other studies have reported shorter-term increases in DOM
(~4-5 years), including an assessment of forest -to-bog restoration of a Scottish lowland raised bog
area, Flanders Moss, where stream water baseline DOM levels were reached within two years at one
site (Shah, 2018). In a Finnish study of the impacts of forest to mire restoration, a short-term peak in
pore water DOM concentration following initial restoration activity was followed by a return to
reference concentrations within six years (Menberu et al., 2017).
Management of peatland for conifer plantation increases DOM concentrations in pore waters and
streams, both during site establishment, potentially during the forest growth, and again as the trees
are felled (by up to 500%) (summarised in Table 3). Forest to bog restoration as a method of land
management produces short-term increases in DOM concentrations while trees are felled and brash
remaining on site decomposes. However, given a long enough timeframe, DOM concentrations appear
to reduce back towards levels seen from comparable control locations. Water companies should note
that this time frame can be up to 20 years in blanket bogs, a time frame considerably longer than the
standard funding cycle.

### 2.4. Managed burning

Managed burning of peatland vegetation (Figure 4) (primarily burning heather for grouse moor
management) is a contentious issue within peatland conservation and management (e.g. Davies et al.,
2016) and has been extensively reviewed over the past decade, particularly in relation to the impacts
on DOM (Worrall et al., 2010; Holden et al., 2012; e.g. Brown et al., 2015), and most recently by Harper
et al. (2018). There is little evidence within these reviews to suggest that DOM concentrations or
colour increase within pore water at the plot scale following managed burns. A recent study showed
no change in DOM concentrations following low and high intensity burning (Grau-Andres et al., 2019),
and in previous studies plot scale DOM concentrations were unchanged (Clay et al., 2009; Clay et al.,
2012; Worrall et al., 2013) or decreased (Worrall et al., 2007a). At the catchment scale it has been
suggested that managed burning contributes to increases in water colour and DOM concentrations
(Clutterbuck and Yallop, 2010; Yallop et al., 2010; Ramchunder et al., 2013). Burning as a management



practice is designed to ensure that there is a mosaic of different aged heather habitat so it seems
plausible that these effects are linked to changes in vegetation cover. As previously discussed *C.*
*vulgaris* produced higher amounts of DOM than *Sphagnum* in the laboratory (Ritson et al., 2016) and
at plot scale (Armstrong et al., 2012). It is also worth noting that Evans et al. (2017b) found that a
wildfire in Northern Ireland resulted in a temporary reduction of DOM concentrations in a
downstream monitoring lake, which was attributed to re-acidification of catchment soils following the
fire.

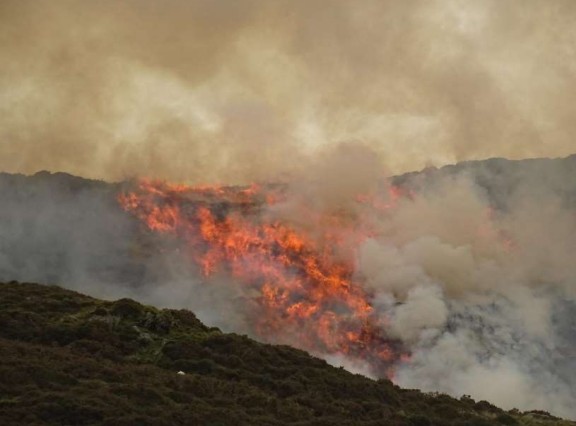


**Figure 4: Burning of vegetation on peat in North Wales (Photo: Chris Evans).**

**Table 3: summary of the published impacts of catchment management activities on DOM concentrations and treatability,**
**focussing on those studies relevant in a UK and Irish context. Numbers in brackets refer to the number of studies showing**
**that effect in each case. Colour coding shows whether the overall conclusion is that effects are positive (green), no /**
**limited change (yellow), or negative (red).**

| Catchment intervention | Impact on DOM concentration | Impact on DOM treatability |
|---|---|---|
| Ditch blocking | **Increase (2)** (Worrall et al., 2007b; Haapalehto et al., 2014)<br>**No change (8)** (O'brien et al., 2008; Gibson et al., 2009; Armstrong et al., 2010; Wilson et al., 2011; Urbanova et al., 2011; Turner et al., 2013; Strack et al., 2015; Evans et al., 2018)<br>**Decrease (5)** (Wallage et al., 2006; Holl et al., 2009; Armstrong et al., 2010; Haapalehto et al., 2014; Menberu et al., 2017) | **No change (5)** (Glatzel et al., 2003; Strack et al., 2015; Gough et al., 2016; Lundin et al., 2017; Peacock et al., 2018) |
| Revegetation to grass species | **Increase (2)** (Qassim et al., 2014; Ritson et al., 2016)<br>**No change (4)** (Parry et al., 2015; Pilkington et al., 2015; Stimson et al., 2017; Alderson et al., 2019) | **Decrease (1)** (Ritson et al., 2016) |



| Revegetation to heather | **Increase (2)** (Armstrong et al., 2012; Ritson et al., 2016)<br>**No change (1)** (Parry et al., 2015) | **Decrease (1)** (Ritson et al., 2016) |
|---|---|---|
| Revegetation to *Sphagnum* | **Decrease (1)** (Armstrong et al., 2012) | **Improve (1)** (Ritson et al., 2016) |
| Forest presence | **Increase (5)** (Cummins and Farrell, 2003; Gough et al., 2012; Gaffney et al., 2018; Howson et al., 2021; Flynn et al., 2022)<br>**No change (2)** (Shah et al., 2021; Flynn et al., 2022) | **Decrease (2)** (Gough et al., 2012; Howson et al., 2021) |
| Clearfell and forest to bog conversion | **Increase (6)** (Cummins and Farrell, 2003; Muller and Tankere-Muller, 2012; Muller et al., 2015; Gaffney et al., 2018; Zheng et al., 2018; Shah and Nisbet, 2019)<br>**No change (3)** (Muller and Tankere-Muller, 2012; Muller et al., 2015; Gaffney et al., 2020) | **Decrease (1)** (Zheng et al., 2018) |
| Managed burning | **Increase (3)** (Clutterbuck and Yallop, 2010, Yallop et al., 2010, Ramchunder et al., 2013)<br>**No change (4)** (Clay et al., 2009; Clay et al., 2012; Worrall et al., 2013; Grau-Andres et al., 2019)<br>**Decrease (1)** (Worrall et al., 2007a) | |



**3: Catchment management impacts on downstream DOM processing**

As indicated by Table 3, there remain considerable knowledge gaps in the area of effects of peatland restoration on raw water DOM concentrations and treatability. This review highlights that both revegetation of bare peat (particularly to *Sphagnum* dominated bog) and ditch blocking have been associated with decreased DOM concentrations within pore waters and ditches at the location restoration occurs. The available evidence also suggests, again at this local scale, that plantation forestry presence and felling tend to lead to increasing DOM concentrations and potentially reduced treatability of exported DOM. However, the evidence for impacts at the stream scale is more equivocal. In the published literature we have been unable to find experimental evidence incorporating local changes in water chemistry in the vicinity of interventions with downstream DOM processing to show whether water quality effects are detectable at the point of abstraction for water treatment works. This extension beyond the plot and hillslope scale represents a significant gap in current understanding, as DOM processing continues within the aquatic environment downstream of peatland restoration sites.

DOM is not conservatively mixed through rivers and lakes but is subject to both biotic and abiotic processing, which change both concentrations and chemical structure (e.g. Tranvik et al., 2009). Loss pathways for DOM include: respiration (Koehler et al., 2012; Stets et al., 2010), sedimentation (Einola et al., 2011; Von Wachenfeldt and Tranvik, 2008), photo-oxidisation (via UV radiation) (Moody et al., 2013; Koehler et al., 2014) and flocculation with naturally-occurring aluminium and iron (Mcknight et



al., 1992; Koehler et al., 2014). DOM is generated within lakes and reservoirs via photosynthesis
(production of algal exudates and release via cell lysis) and through processing of particulate matter
(Tranvik et al., 2009) so that DOM concentrations at the point of abstraction from reservoirs represent
the sum of these removal and generation processes.
DOM produced via these processes is relatively transparent and hydrophilic in comparison with DOM
generated by organic rich soils, and thus presents different challenges for treatment, particularly as
the hydrophilic DOM is not easily removed through coagulation (Matilainen et al., 2010) and may lead
to the need for additional capital investment in order to effectively reduce residual DOM in drinking
water.
Importantly, in-reservoir algal production, and hence within-reservoir generation of DOM, is often
limited by the availability of phosphorus, nitrogen or both. Hence, waterbodies with high
concentrations of inorganic nutrients, either delivered externally from their catchments or re-released
internally from sediments, are likely to generate additional DOM within the water column
(Feuchtmayr et al., 2019; Evans et al., 2017a). Further, evidence is growing on the importance of lake
and reservoir bed sediments as a direct source of DOM to the water column, with reducing conditions
occurring during stratification of lakes and reservoirs causing redissolution of previously sedimented
organic matter (Peter et al., 2017).
In their assessment of DOM in lake inflows and outflows, including those of several reservoirs, Evans
et al. (2017a) concluded that any measures that can reduce N and P export from the catchment or
release from sediments, or which can strip nutrients from the water column, could provide effective
mitigation for high DOM concentrations by reducing algal DOM production. For example, measures
for reducing nutrient loading to lakes from the catchment (Spears and May, 2015) and bed sediments
(Spears et al., 2016) can be effective in reducing algal biomass in UK lakes - although the effects on
algal DOM production in relation to drinking water treatment require further assessment. To date,
this option has rarely been considered in relation to DOM-related treatment issues, although nutrient
management is often considered in relation to other (taste and odour) related treatment issues. The
available evidence therefore suggests that measures to reduce taste and odour problems could deliver
co-benefits in relation to DOM levels.
A future research focus should therefore include answering the question of whether measures which
reduce in-reservoir DOM production, and/or favour in-reservoir DOM removal, may be as – or perhaps
more – effective than measures aimed at reducing DOM export from the terrestrial catchment.  For
lakes acting as DOM sources, management regimes that reduce nutrient (primarily N and P) inputs
from catchments and/or internal loading of nutrients and DOM from sediment to the water column
may be more effective than those focussed on reducing inflowing DOM concentrations directly.
Restricting nutrient inputs is also likely to reduce organic nitrogen concentrations relative to organic
carbon concentrations, which has the added benefit of reducing the formation potential of
nitrogenous DBPs. In addition, Birk et al. (2020) suggest that rising DOM loading from the catchment
may act to dampen algal responses to nutrients through light limitation of primary production within
some European lakes. If, by extension, this also limits in–reservoir DOM production then catchment
interventions that relieve DOM load, but not nutrient load, may result in an increase in in-reservoir
DOM production. Even in the case of less nutrient-rich water bodies, it appears that reducing N and P
loadings would be beneficial for water treatment as this is likely to restrict additional DOM formation.




## 4. Conclusions

Increasing DOM concentrations in reservoirs draining catchments dominated by peat soils are a cause for concern for water companies, from both regulatory compliance and treatment cost perspectives. To a large extent this increase appears to be a long-term large-scale phenomenon, driven by improvements in air quality, and thus beyond the direct control of catchment managers. While it is likely that atmospheric deposition-driven changes in DOM are beginning to level off it is also feasible that future climate change could also contribute to further increases in concentrations. The production of DOM in peat soils, for example, is known to be highly sensitive to soil temperature (Clark et al., 2009) while long-term increases in precipitation have also been linked with DOM increases (De Wit et al., 2021).

To date, catchment management initiatives, while providing clear overall restoration benefits for peatlands, do not appear to have produced a generalised solution to the challenge of stabilising or reversing DOM increases in drinking water sources, although there is some evidence that catchment interventions may provide benefits for DOM export in specific cases. We have identified some areas where there is mounting evidence for the importance of certain catchment interventions. In particular, short-term effects of forest felling and harvesting activities have repeatedly shown to have detrimental effects on DOM concentrations. Catchment interventions may also provide co-benefits such as reductions in sediment and particulate organic carbon loadings to reservoirs, reductions in greenhouse gas emissions and enhancement of biodiversity, which may justify the implementation of measures when all benefits are combined, even if the direct benefits for DOM alone may not.

Our review of the published literature highlights a major current evidence gap of importance to the water industry: the quantification of the impacts of catchment management on DOM concentration and treatability at the point of abstraction. The size of the research challenge with respect to the necessary spatial and temporal scale and need for robust Before-After-Control Impact (BACI) of any field experiment cannot be underestimated, and perhaps explains in part the current dearth of reliable information. This is particularly pertinent when changes in water chemistry may take a number of years to be seen, depending on catchment dynamics and within reservoir processes. Our review has highlighted that in-reservoir biogeochemical processes should be considered alongside catchment land management approaches by the water industry to maximise the potential for upstream solutions to rising DOM concentrations in source waters.

Catchment management measures that reduce in-reservoir DOM production, or favour in-reservoir DOM removal, may be as or more effective, particularly with respect to more nutrient rich systems. More generally, it seems clear that catchment management should be considered part of the response strategy to rising DOM levels, and as part of a process to improve the resilience of source waters, not a panacea. It is therefore important that the water industry also develops effective tools to predict likely future DOM levels resulting from a combination of large-scale and catchment-scale drivers, to ensure that investments in both catchment management measures and DOM treatment infrastructure are correctly targeted, integrated, timely and cost-effective.

The authors declare that they have no conflict of interests.

The work was conceptualised by DM and CE, funding acquisition was by DM, CE and BS, JW carried out the initial review and wrote the manuscript with contribution from all authors.




Acknowledgements:
We thank staff from Scottish Water, United Utilities, Yorkshire Water, Irish Water and Dŵr Cymru
Welsh Water for their informative discussions and comments on early drafts of this manuscript.
Discussions with Nadeem Shah and Tom Nisbet regarding work being undertaken by Forest Research
are also gratefully acknowledged, as are the comments from 2 reviewers of an earlier draft of this
manuscript. This work was funded by a NERC Environmental Risks to Infrastructure
Innovation Programme grant NE/R009198/1, UKRI SPF UK Climate Resilience programme – Project no.
NE/S016937/2, NERC LTSM LOCATE (Land Ocean Carbon Transfer, NE/N018087/1) and Scottish Water.





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
