# Peer review of "Reviews and Syntheses: Understanding the impacts of peatland catchment"

_Biogeosciences, 2022_

## Author Comment (AC2)

*We thank the reviewers for their helpful and constructive comments that have enabled us to improve the manuscript. We have responded to each comment in the text – the reviewers comments are in plain text, while our responses are in italics.*

**Reviewer 1:**

General comments

This review paper pulls together studies on the effect of peatland restauration measures on DOM quantity and quality on receiving waters. It states a very diverse response to these measures and calls for more experimental studies. The paper matches the scope of Biogeosciences.

While I very much appreciate the overview on this body of literature I have major concerns regarding the resulting manuscript. For me parts of the abstract and the introduction are too long and detailed while other aspects are underrepresented. The study seems to have a UK focus but sometimes takes in international papers. It remains unclear to what extent the results and conclusions can be transferred to other places. I major shortcoming is the lack of discussion on the timescale of the reviewed studies and on the role of time in water quality response to the measures. Finally, references in the text are sometimes sparse and not all statements are underpinned with literature.

*Thank you for your comments on the manuscript. The review aimed to focus on papers that would be relevant to upland catchment management, primarily in a UK and Irish context. We have checked the manuscript and updated and added references, with the detailed information supplied in answer to the comments below.*

I hope my specific comments below will help to improve this manuscript.

Specific comments

Abstract

1. For me the abstract is too long and too detailed. e.g. L16-23 can be greatly condensed to introduce the problem to be solved.

*We have condensed the abstract, especially the section highlighted by the reviewer and it now reads as follows:*

*"In the UK, most large reservoirs constructed for public water supply are in upland areas and situated in catchments characterised by organic-rich soils including peatlands, and*

*often considered to be in sub-substandard condition. Such catchments leach large amounts of dissolved organic matter (DOM) to water, with water draining peatlands tending to release the most. High and rising DOM concentrations in these regions raise treatment challenges for the water industry.*

*In the UK, water companies are increasingly considering whether upland catchment peat restoration measures can to slow down or even reverse rising source water DOM concentrations and thus reduce the need for more costly and complex engineering solutions. There remains considerable uncertainty around the efficacy of such measures, and a comprehensive overview of the research in this area remains lacking. Here we review the peer-reviewed evidence for the effectiveness of four catchment management options in controlling DOM release from peat soils: ditch blocking, revegetation, reducing forest cover, and cessation of managed burning.*

*Results of plot scale investigations into effects of ditch blocking on DOM leaching are currently largely equivocal, while there is a paucity of information regarding impacts at spatial scales of more direct relevance to water managers. There is some, although limited evidence that terrestrial vegetation type may influence DOM concentrations and treatability. The presence of plantation forestry on peat soils is generally associated with elevated DOM concentrations, although reducing forest cover has little short-term benefit and can even exacerbate concentrations further.*

*Catchment management measures have rarely been monitored with downstream water quality as the focus. To mitigate the uncertainty surrounding restoration effects on DOM, measures should be undertaken on a site-specific basis, where the scale, effect size and duration of the intervention are considered in relation to subsequent biogeochemical processing that occurs in the reservoir, the treatment capacity of the water treatment works and future projected DOM trends."*

2. L29-28: It would be easier to read the result part if the names of the four management options listed above would show up here explicitly. Rather details (e.g. sentence with Sphagnum) blur the picture.

*We have reworded this sentence as follows:*

*"Results of plot scale investigations into effects of ditch blocking on DOM leaching are currently largely equivocal, while there is a paucity of information regarding impacts at spatial scales of more direct relevance to water managers. There is some, although limited evidence that terrestrial vegetation type may influence DOM concentrations and treatability. The presence of plantation forestry on peat soils is generally associated with elevated DOM concentrations, although reducing forest cover has little short-term benefit and can even exacerbate concentrations further. "*

Chapter 1: Introduction

3. L48f: This first statement would profit from a reference to the Net Zero strategy.

*We have included a reference to the UK Government's Net Zero strategy.*

4. L53f: Wording - "tend to be relatively high". Relative to what? Why "tend"?

*We have modified this sentence as follows to clarify that we are referring to high concentrations of DOM from peatlands:*

*"Peatlands release particularly high amounts of organic matter as dissolved organic matter (DOM) into drainage waters, and DOM concentrations have been rising since the 1980s (e.g. Naden and Mcdonald, 1989; Robson and Neal, 1996; Harriman et al., 2001; Freeman et al., 2001; Worrall et al., 2004)."*

5. L61f: It would be helpful to state that there are diverging views on the cause of DOM concentration increases and surely more than two references are needed.

*We have included reference to a recent paper (Monteith et al 2023) showing that declining ionic strength is the only variable that consistently seems able to produce the DOM trends in time and space that have been seen across Europe and North America. We have included reference to other causes of DOM concentration increases and the conclusion from Monteith et al (2023) that these will become increasingly important once trends in declining ionic strength stabilise. While we understand the reviewer's concern that there are differing views on the causes of the DOM increase in surface waters the main point of the paper is whether managing upland peat dominated catchments can stabilise or reverse the increasing trend that has been seen so we don't want to overly lengthen the introduction discussing the differing views on the potential causes of the increase.*

6. L67f: Is there a reference for this last statement or are you the first one systematically asking the question on the impact of restoration on water quality?

*The work initially aimed to review the impact of peatland restoration activities on DOM concentrations at reservoir outlets. We found no published literature focussing on these downstream effects – all studies published focussed on the impacts at a spatial scale more local to the interventions being carried out. Following the reviewer's query we have rewritten this sentence as follows:*

*"It has been proposed that peatland restoration measures might help slow or even reverse these DOM trends, along with other important benefits including increased terrestrial carbon storage, water retention and improvements in upland biodiversity (e.g. Glenk and Martin-Ortega, 2018)."*

7. Figure 1: I would expect a statement of the number of sites being part of that plot.

*We have included the number of sites in the legend (23 sites).*

8. L79: No need to define an abbreviation for DWI when this term is not used in the manuscript a second time.

*We have removed the abbreviation, thank you. Other references to the DWI have since been removed.*

9. L78-89: This is way too long and detailed for a paper with a focus on management options and not on the chemistry behind disinfection byproducts.

*We have removed the section on disinfection methods (lines 79 -84) while retaining the information on the difficulties DOM causes the water industry.*

10. L90-103: Is DOM composition a point later on in the result section? If not, try to boil this down to the information necessary to understand the rest of the manuscript.

*We do cover the impacts of catchment management on DOM composition and hence treatability where this information has been available in the literature. However, our searches did not find many papers focussing on this element. We suggest keeping the information on DOM composition due to its relevance for drinking water treatment.*

11. L108f: What is the time reference of costs? Overall, per year? For this and the following statement a reference is needed!

*These are communications from the water industry partners in the project so we have referenced the fact sheets produced during the sector engagement meetings within the project. We have amended the sentence slightly to remove direct comments on costs – these were initially covered in discussion with water industry partners. It now reads as follows:*

*"Major additional costs are incurred where capital investment is needed to upgrade treatment infrastructure designed for lower concentration ranges experienced in the past (Monteith et al., 2021)."*

12. L112ff: Here "peatland restauration" kicks in a bit surprisingly. Why only looking at peatland restauration? Yes, mentioned in the first sentence of the introduction. However, the definition of peatland as the dominant source of DOM in headwaters was not that clearly done so far. This needs a better

connection. It is totally fine to limit the review to peatland restauration but it needs to be justified and clearly narrowed down in the introduction section.

*We have included the following text as the closing paragraph of the introduction to clearly signpost the reader that the rest of the review will focus on peatland restoration:*

*"Peatland restoration, i.e. physical interventions to return areas of the uplands to a more natural state (i.e. high water table and plants adapted to wet environments) has been suggested as a catchment scale method for reducing DOM concentrations in water draining peatlands (IUCN Peatland Programme). The primary restoration methods undertaken to date in the UK uplands are: blocking of peatland drainage to raise the water table, revegetation of bare peat with peatland species, removal of plantation forestry to allow peatland species to recolonise and water tables to rise, and cessation of managed burning to encourage growth of peatland plant species (Figure 2) (IUCN Peatland Programme). It is important, therefore, for water industry decision makers to understand the extent to which peatland restoration could make a positive contribution to reducing DOM concentrations of raw water and thus relieve stresses on the treatment system and potentially remove the need for major additional capital investment in treatment plant. This work reviews the available peer-reviewed literature and provides an assessment of the impacts of peatland restoration on DOM concentrations and treatability."*

*We have also added the following paragraph to provide methodological information and link the introduction to the remaining sections more clearly:*

*"To answer the question "will peatland catchment management reduce DOM concentrations in raw water" we explored the evidence within the peer-reviewed scientific literature for catchment management approaches within peatland dominated drinking water catchments to influence DOM concentrations in the soils and waters of peatland catchments. This was achieved by applying a standard set of Boolean search terms within Web of Science and Google Scholar. The terms were: ("dissolved organic matter" OR "dissolved organic carbon" OR "DOM" OR "DOC" OR "colour") AND ("peatland" OR "bog" OR "fen" OR "moor") AND ("ditch blocking" OR "forest" OR "plantation" OR "managed burning"). Initial results, including titles and abstracts, were rapidly reviewed to determine whether the information within the papers was relevant, both in terms of subject matter and in region (limited to temperate peatlands), then relevant papers were read in full and included in the review."*

*We have also made a minor revision to the opening paragraph, to make the link between reservoirs containing water from peatlands and this water being high in DOM: "Nearly three quarters of the storage capacity of drinking water reservoirs in the UK is sourced*

*from peatland areas (Xu et al., 2018). The dissolved organic matter (DOM) concentrations of water from draining from peatlands are high, and have been rising since the 1980s (e.g. Naden and Mcdonald, 1989; Robson and Neal, 1996; Harriman et al., 2001; Freeman et al., 2001; Worrall et al., 2004)."*

13. L116f: Why stating "qualitative" here? Because literature does not allow a quantitative view? The latter would be much more helpful for the water industry, right? Wouldn't it be rather a result and conclusion than an objective that only qualitative but not quantitative assessment is possible with the given literature? I miss the statement on the four management options that are presented in the abstract.

*We have changed the wording of the abstract to the following:*

*"Here we review the peer-reviewed evidence for the effectiveness of four catchment management options for peat soils: ditch blocking, revegetation, reducing forest cover, and cessation of managed burning."*

*Qualitative was stated here because there was insufficient evidence to be able to say management option X reduces DOM concentrations by Y+/-z. However, we understand your point that this is more of a concluding statement than an opening one and suggest removing the term qualitative.*

Chapter 2: Evidence for the efficacy…

14. L124: You jump in directly with ditch blocking without introducing earlier why this specific measure may help DOM reduction

*We have included the following sentences at the end of the introduction, after "in the past." on line 112, to introduce the measures and why they may work:*

*"Peatland restoration, i.e. physical interventions to return areas of the uplands to a more natural state (i.e. high water table and plants adapted to wet environments) has been suggested as a catchment scale method for reducing DOM concentrations in water draining peatlands (IUCN Peatland Programme). The primary restoration methods undertaken to date in the UK uplands are: blocking of peatland drainage to raise the water table, revegetation of bare peat with peatland species, removal of plantation forestry to allow peatland species to recolonise and water tables to rise, and cessation of managed burning to encourage growth of peatland plant species (Figure 2) (IUCN Peatland Programme)."*

15. Fig. 2: Homogenize style of the different text elements. E.g. capitalized letters or not, colon or not… Right part of the figure on the water treatment is too

small to be really helpful and may need a reference for the figures source. Strange to see, again, the catchment and reservoir in this part of the figure. Figure 2 seems not to be referenced in the manuscript text and also does not clearly follow the structure of chapter 2. Where, e.g., is ditch blocking in the figure?

*We will check the style of the figure sections. We will highlight the restoration actions shown in the figure and refer to the figure in the relevant sections. The section indicating the water treatment works was designed for illustrative purposes and not to demonstrate any particular treatment type.*

16. L128-130: This statement needs a reference.

*We have included a reference to the IUCN Peatland Programme website as this has numerous examples of where, why and who is undertaking peatland restoration.*

17. L133: It would make sense to state the temporal scale of this reduction quantity. There can be initial effects and a longer-term evolution of concentrations. This is also true for the section L138-143. This information is in Tab. 1 but not in the text!

*We have included the range of timeframes in the text on line 133, and in the section covering L138-143. This now reads as follows:*

*"The studies investigated effects between five and twenty years following ditch blocking, and reported a cross-study average 34% reduction in DOC concentration (range 0 to 69%) (Wallage et al., 2006; Holl et al., 2009; Haapalehto et al., 2014; Strack et al., 2015; Menberu et al., 2017)."*

18. Tab. 1: In this style I suggest to move the table to the supplement. Maybe rather condensed and/ or visual information could be shown? e.g. as a bars with % increase/ decrease on y and time since blocking on x? I do not insist here, acknowledging the point that studies are hard to compare. However, the table is hard to grasp for the reader as information is hidden in text sections within the table.

*We thank the reviewer for the suggestion that the information would be easier to assimilate in graphical form, and suggest the addition of a set of graphs covering pore waters and streams as parts a) and b) of a figure if that can be incorporated at this stage.*

19. L179-181: You mention "suitable timescales" of observation but do not define them nor introduce them earlier on in the introduction section. However, this is obviously a relevant point that needs more scientific background earlier on.

*This point is interesting and we agree relevant. However, as part of the review we looked for papers monitoring the impacts of restoration on DOM concentrations at reservoir outflows and were unable to find any studies. We do know from other long-term monitoring studies, such as the Upland Waters Monitoring Network, that detecting long term changes from the noise of inter-annual variability needs those decadal or longer datasets.*

*In order to keep the focus of the review on the impacts of catchment management on DOM concentrations we have removed the part of the sentence referring to timescales, instead ending the sentence after "hugely challenging logistically and financially to design and maintain."*

20. L188-192: These statements needs references.

*We have added references to the Pilkington et al 2015 report as this explains the issues and restoration efforts in the Peak District region to date.*

21. L193f: If research is limited on what basis was the decision made that re-vegetation needs an own chapter? I am not in doubt here but the manuscript does not explain that to the reader.

*We have included the rationale for the inclusion of the sections in the introduction (see response to point 12).*

22. L206: Check consistent spelling of re-vegetation.

*We have checked with Collins and Merriam-Webster dictionaries and both have revegetation as an acceptable spelling. We have changed the heading of section 2.2 to match the text.*

23. L235f: Given the large body on literature on forestry-water quantity and -quality issues I suggest to cite rather review-style papers here than this very narrow selection.

*We have checked for review papers on the subject but the aim of this review was to include only those papers which had incorporated monitoring DOM concentrations and/or fluxes from forestry on deep peat that was directly relevant to the UK and Irish context where blanket bogs in the uplands are generally treeless in their natural state.*

24. Table 2: This table is not embedded in the text of the matching chapter 2.3 (between lines 233 and 258). Which information is given in the text and which in the table? Compared to Table 1 there is no temporal reverence on the before-after comparison. Is there a reason why this table focus on UK only but Table 1 have a more international width? Other than that I suggest to consider, similar to Table 1 alternative forms to display information.

*In the UK the majority of plantation forestry is on blanket bog or raised bog that would not naturally have any tree cover. More continental bogs and fens have at least some tree cover as part of the natural vegetation so this section is limited to the UK and Irish context to ensure that the information is relevant. We have included a column showing the timeframe of the monitoring and again mentioned in the relevant text (similar to the comment on the ditch blocking section).*

*I am not sure what the reviewer means by embedding the table in the text – is this referring to where in the review it is placed or part of the PDF formatting?*

25. Chapter 2.3 as a whole: I have problems following the logic of this chapter and the switch of topics from section to section. This makes it hard to see the bigger picture of knowledge on forestry and DOM.

*We have included an additional sentence at the start of the section setting out how the section is ordered:*

*"This review covers the impact of ground preparation and forest planting, in-situ forest growth, and forest removal (including forest to bog restoration) on peat)"*

26. Table 3: I like the idea of the color scheme here. The type of catchment intervention does only parly match the structure of the manuscript so far. This needs improvement.

*Thank you for your positive comments on the colour scheme. We separated out the revegetation to different peatland species in the table as the available data suggests that the outcomes were different. However, we appreciate that this means that the table does not as closely match the section headings. We have changed the section headings of these rows to be revegetation and then the specific vegetation change in brackets. We consider that the potential for different effects from the growth of different vegetation species is important enough to justify the additional rows in the table.*

Chapter 3: Catchment management impacts

27. The first part of this chapter seems to be a discussion and interpretation on missing knowledge. However, it seamlessly propagates into another result section on DOM processing in the catchment. Some parts of the studies reviewed before already contains instream processes as they have been observed in stream. This drawing of a line between upstream and downstream processes is totally unclear for the readers. After this part the text comes back to a recommendation on future research… All this needs a clearer structure.

*We have changed the title and structure of the last part of the review in response to the reviewer's concerns. It is now titled as discussion and conclusion and is structured as follows:*

- *Discussion of results and interpretation of current knowledge*
- *Possible reasons for the evidence gap of the effects of catchment management on DOM downstream*
- *Concluding paragraph*

Chapter 4: Conclusions

28. L425-429: In this concluding section no new knowledge should be introduced. This is rather part of the introduction.

*Lines 427 – 429 were copied in error, we will remove these and thus the part covering new knowledge that was inadvertently introduced.*

**Reviewer 2:**

I enjoyed reading this docuement, and most of my comments are easily dealt with. I have included some detailed comments on an attached version of the manuscript.

My major comments is that the paper needs a Methodology. Literature review is not just an essay on a subject, it has a an aim, a question and a method. The lack of a method means:

- there a inconsistencies in what is or is not included in the review - see the abstract where some things are listed but not then commented on.

- there are inconsistencies in the scope of the literature reviewed - I have already assumed that only literature from peer-reviewed literature is included although that is never stated - for example in some sections UK data is used; in others it is from outside the UK and in still others it is only British Isles data.

- how was data combined? In some sections there are median effects and in others there are mean difference. Some sections talk of signiticant effects and others seem to be mean important by significant.

- lack of clearly defined method and aim means that snippets of method and discussion appear in the results

The conclusion is not a conclusion, there seems to be a better conclusion in the discussion. Whole new ideas appear in the conclusion that were never mentionned in the rest of the text.

There are other reviews on related subjects that do use an appropriate meta-analysis for their review and so I do think the authors should have a look at some other reviews and meta-analyses to see how to structure their paper.

At the moment the paper reads like a small report given to industry partners and not a literature review for an international journal.

*We thank the reviewer for their comments on the manuscript although unfortunately we have been unable to access the additional comments the reviewer mentions. To address the major comment above regarding the methodology of the literature review, this work was conceived as a literature review rather than a systematic review. We have included a paragraph on the literature search methodology to highlight the question to be answered and the methods in the review. This could be worded as followed:*

*"To answer the question "will peatland catchment management reduce DOM concentrations in raw water" we explored the evidence within the peer-reviewed scientific literature for catchment management approaches within peatland dominated drinking*

*water catchments to influence DOM concentrations in the soils and waters of peatland catchments. This was achieved by applying a standard set of Boolean search terms within Web of Science and Google Scholar. The terms were: ("dissolved organic matter" OR "dissolved organic carbon" OR "DOM" OR "DOC" OR "colour") AND ("peatland" OR "bog" OR "fen" OR "moor") AND ("ditch blocking" OR "forest" OR "plantation" OR "managed burning"). Initial results, including titles and abstracts, were rapidly reviewed to determine whether the information within the papers was relevant, both in terms of subject matter and in region (limited to temperate peatlands), then relevant papers were read in full and included in the review."*

More specific comments:

 - remember to include the subject of the sentence.

 - there are plenty of pregnant sentences where a reference is expected but not given.

*We have included additional references in the text where highlighted by reviewer 1.*

---

## Author Response (AR2)

General comments
I was reviewer in the first round and like the idea of this manuscript. The evident lack of consistent studies on the effect of measures in the catchments is something to be brought to the biogeochemical community. I think the paper is much clearer now and the authors carefully addressed the raised points. I still have some suggestions. My major point is that half of the discussion chapter is addressing DOM production in surface waters and nutrients N&P that was not at all part of the result chapter. Two things follow for me: I miss a brief overview on processes known for water fluxes and pathways and DOM sources, mobilization and fate in peatland catchments accompanying Fig. 2. That can be really brief but would help the reader to better understand the effect of measures. This may also include the production of DOM in the surface water and therefore allow to shift part of the discussion to that. In general I miss some consistency in the terms used for measures in peatland catchments - the manuscript offers quite a variety here. Finally, I don't like that the methodology is part of the result chapter.
For more details and minor things refer to my specific comments below.
*We thank the reviewers for their comments and feedback, they are much appreciated. We have added an opening paragraph on DOM production and fate in the opening paragraph to allow the discussion to provide more detail on this. We also suggest removing the original opening paragraph as the information contained within it is covered elsewhere in the introduction and the information flows better without it being there. In addition, we have suggested that the catchment schematic figure becomes Figure 1 so that the overall picture of the areas covered is clear at the start of the review.*
*Answers to specific points are written in italics in the text below.*

Specific comments
Abstract
L25-26: Do you use effectiveness and efficacy as synonyms? Not clear for me.
*We have changed the use of efficacy to effectiveness*

Introduction
L76: I cannot find this reference in the reference list. Is that focused on UK as the mentioning of the Drinking Water Directorate suggests? Please check references in general – in times of reference managers I cannot imagine how a reference can vanish from the list.
*We are not sure how Endnote lost the reference – we have added it in and have checked all others, thanks for spotting. The reference refers to the potential carcinogenic nature of THMs and other DBPs.*

L94-101: This is a nice section but I am not sure if this explanation is needed for the reader as you already mentioned the same fact in line 59-61.
*We feel that the two sections are different enough for the explanation to remain – lines 59-61 are referring to the reason for DOM concentration increases in raw water, while the later section is covering what the impact is on the water companies treating the water. We think the latter is important to remember especially as water treatment companies are in the news at present for their impacts on downstream water quality but the wider impacts of spending priorities on human health from a drinking water perspective should be remembered too.*

L109: "understanding the extent" refers to a quantification, right? Totally fine. What about understanding processes that lead to the changes in DOM concentrations? If that is an aim here, it should be mentioned.
*It wasn't a main aim – the overarching aim of the literature review was originally to demonstrate the what rather than the why -so to answer the questions whether the peatland management would change DOM concentrations not to review the processes behind that change.*

L112f: I think the objectives cannot be given in one single sentence only. This needs more focus and detail: Spatial and temporal scale? UK only or beyond? What is meant with "impacts" – concentration, flux, quality?
*We have added more information to the objectives as follows:  In this study, we review the available peer-reviewed literature relating to the impacts of peatland restoration on DOM concentrations and treatability of raw drinking water.  Finally, we consider the possible influence of catchment land-use on in-reservoir DOM cycling, and what impact this may have had on drinking water treatability.  We focus on the UK as a well-studied area in which peatlands make an important contribution to drinking water supplies, and where rising DOM concentrations are having a significant impact on water treatment processes and costs, but the conclusions of the work are likely be relevant to other areas with peat-derived water supplies.*

Evidence…
The first section is a method and not a result section. With some additional information as requested below I would rather make it an own chapter than a part of the results. I moreover wonder if there was any filter applied for the country and continent the papers are referring to.
*We have changed the layout so that the methods paragraph sits within its own methods section and added the information on the peatland regions prioritised in the review. The filter applied on first search was to keep results from temperate peatlands and to include those from outside the UK and Ireland where there was insufficient evidence in the UK and Irish contexts. We have added the following sentence to clarify the spatial scale: Given the geographic focus of the project, we prioritised papers from the UK and Ireland where available, but also drew on data from other temperate peatland regions where required.*

L117: This question is not fully similar to the objective and somewhat new. Is peatland catchment management similar to peatland restoration?
*Yes, this is covering the same, we have changed the wording to match the introduction.*

L124f: For me it would make sense to state the numbers of papers considered for the different steps and how much have been finally considered. Maybe this could be enlarged to the countries/ continents the studies are referring to.
*We have added the following information to demonstrate the numbers of papers included in the review: From the original searches, 272 papers were considered relevant enough for further reading and 104 were included in the review.*

L129-131: These statements need a reference. Or is that all covered by Clark et al in the next sentence?
*We have added reference to Holden et al 2011 – who measured water table depths in drained, rewetted and intact areas of peatland*

L134: Is conservation here used similar to restoration? If yes, I suggest to avoid using different terms for the same thing. If yes, please define the term here.
*We have changed conservation to restoration for consistency.*

L137: Consider to adjust this sentence. I had to read it several times to get it.
*We have rewritten as follows: …DOM concentrations had been assessed in pore waters, in ditches and in streams….*

L139: Is peat soil water the same as pore waters in the sentence before?
*Yes, changed terminology for consistency.*

L146ff: Would be interesting to know if some of these 11 studies are also mentioned in the 5 studies that found the decrease in pore water. That would better underline the statement in L144.
*As mentioned in Table 1 only one measured both and the changes were not consistent.*

Table 1: The meaning of "chronosequence" and "survey" as a study design is not clear from the table description.
*We have added some additional text to improve clarity as follows: Reference to chronosequence in the survey design refers to a sampling strategy whereby sites that had had interventions at different times were used as a proxy for control sites, while survey refers to a short term one-off sampling of multiple locations.*

L231: Why is there a reference to Table 3 here that is actually describing forestry activities?
*The reference to Table 3 refers to the table summarising all the restoration actions. I think the table legend is on the previous page with the current formatting.*

L248-261: This is the first time you actively mention studies outside UK. As written above this should be already mentioned in the objectives and methods.
*The initial premise of the review was to focus on UK and Irish specific studies where there was evidence available as these would be the most relevant to UK and Irish water companies. However, as the initial searches included all temperate peatlands where there was potentially relevant evidence from outside UK / Ireland this was included rather than saying that there was no evidence available. We have clarified the approach used in the methods section.*

L311ff: I have problems following here. What is "such restoration" referring to? Forest to bog with tree to waste? Would be helpful to make that clear here.
*We were referring to forest to bog restoration – have changed the wording to clarify.*

L333: The reference to fig. 2 is not clear for me.
*The reference to Figure 2 refers to the managed burning shown in the whole catchment schematic. Note that this has been changed to Figure 1 (see above)*

Table 3: Another term for the (I assume that) same thing: catchment intervention. The term "clearfell" is not used in the manuscript and not fully clear for me.
*We have changed tree felling on line 276 to clear felling to clarify that we are talking about the large scale felling of trees across an area not individual tree removal. Catchment intervention in Table 3 was used because it provides an overarching term that covers human interventions in the drinking water catchment without a-priori stating whether they are positive or negative. The other option could be catchment management.*

Discussion
L362f: Is there a review paper you can cite here for the other positive effects?
*Added a recent reference from Loisel and Gallego-Sala (2022) covering the ecological resilience of restored peatlands.*

L394-444: I have issues that a completely new point is raised here. Until this section the manuscript was about terrestrial sources and mobilization but not about DOM production in the surface water and N and P. This hints to a missing overview on DOM sources, pathways and budgets in peatland catchments in general that this review may provide as an overview in the starting chapters. This section could be integrated there as well as this is not based on a result of this manuscript. Overall the weight of this section is too large in the discussion compared to the preceding part that is based on the reviewed papers in the result section.
*We have included an opening overview of DOM sources, pathways and budgets as the opening to the manuscript to highlight that the DOM that is in the water at the point of abstraction is not necessarily the same as the DOM that is released from the peat in the catchment (irrespective of peatland restoration). We have also signposted this section in the final paragraph of the introduction as a "this needs to be considered in future when considering catchment management as an approach to improve water quality and when considering the design of studies at a catchment scale". We have separated the discussion into sections to highlight this approach. We have also reduced the length and detail of the section to reduce the weight of this compared to previous.*

L444: Interesting that the water industry should develop the tools. I would enlarge that a bit – science obviously needs to provide underlying process knowledge. I see it then more of a joint task of scientists and practitioners to provide more monitoring-based evidence on the one hand and to develop predictive models on the other hand.
*We agree and have expanded the sentence to say that research science and the water industry need to work together to develop the monitoring and modelling needed.*